# Systematic Review of Pain in Clinical Practice Guidelines for Management of COPD: A Case for Including Chronic Pain?

**DOI:** 10.3390/healthcare7010015

**Published:** 2019-01-22

**Authors:** Hayley Lewthwaite, Georgia Williams, Katherine L. Baldock, Marie T. Williams

**Affiliations:** 1Alliance for Research in Exercise, Nutrition and Activity, School of Health Sciences, Division of Health Sciences, University of South Australia, Adelaide 5001, Australia; Georgia.Williams2@sa.gov.au (G.W.); Marie.Williams@unisa.edu.au (M.T.W.); 2Australian Centre for Precision Health, School of Health Sciences, Division of Health Sciences, University of South Australia, Adelaide 5001, Australia; Katherine.Baldock@unisa.edu.au

**Keywords:** chronic obstructive pulmonary disease, pain, chronic pain, symptom palliation, clinical guideline

## Abstract

Chronic pain is highly prevalent and more common in people with chronic obstructive pulmonary disease (COPD) than people of similar age/sex in the general population. This systematic review aimed to describe how frequently and in which contexts pain is considered in the clinical practice guidelines (CPGs) for the broad management of COPD. Databases (Medline, Scopus, CiNAHL, EMbase, and clinical guideline) and websites were searched to identify current versions of COPD CPGs published in any language since 2006. Data on the frequency, context, and specific recommendations or strategies for the assessment or management of pain were extracted, collated, and reported descriptively. Of the 41 CPGs (English *n* = 20) reviewed, 16 (39%) did not mention pain. Within the remaining 25 CPGs, pain was mentioned 67 times (ranging from 1 to 10 mentions in a single CPG). The most frequent contexts for mentioning pain were as a potential side effect of specific pharmacotherapies (22 mentions in 13 CPGs), as part of differential diagnosis (14 mentions in 10 CPGs), and end of life or palliative care management (7 mentions in 6 CPGs). In people with COPD, chronic pain is common; adversely impacts quality of life, mood, breathlessness, and participation in activities of daily living; and warrants consideration within CPGs for COPD.

## 1. Introduction

In people with chronic obstructive pulmonary disease (COPD), persistent pain is a common clinical issue adversely impacting daily function, symptom burden, and quality of life [1,2,3]. To date, there are at least 34 studies [1,4,5,6,7,8,9,10,11,12,13,14,15,16,17,18,19,20,21,22,23,24,25,26,27,28,29,30,31,32,33,34,35,36] and two systematic reviews [2,3] available that report direct estimates of pain prevalence in people with COPD (Figure 1). Prevalence rates for pain vary markedly and range between 21 to 82% (Figure 1) with a mean pooled estimate of 66% (95% Confidence Interval (CI) 44–85%) [2]. These estimates are likely influenced by the clinical context of the sample (e.g., stable, acutely unwell, or end of life), sampling source (e.g., outpatients, pulmonary rehabilitation, and population), assessment instrument, and focal period (e.g., past week, month, and year). Excluding studies where the prevalence is likely to reflect acute pain associated with hospital presentation or end of life care, the majority of studies reflect persistent or chronic pain reported by people participating in pulmonary rehabilitation, attending routine outpatient appointments, or of the broader community. Compared to the general community, large population-based studies report higher rates of persistent pain in people with COPD for (1) “usual” chronic non-cancer pain (34.9% COPD versus 14% control; odds ratio (OR) 1.56, 95%CI 1.40–1.74) [29]; (2) chronic low back pain over 12 months (44.8% COPD versus 28.4% control; OR 1.38, 95%CI 1.16–1.64) [33]; (3) chronic neck pain over 12 months (40.5% COPD versus 26.1% control; OR 1.21, 95%CI 1.02–1.45) [33]; and (4) chronic pain lasting longer than six months (51.8% COPD versus 25.9% control; OR 1.56, 95%CI 1.30–1.92) [34]. People with COPD who have persistent pain have been reported to have significantly higher pain medication use over a 12 month period and higher usage of pain-related services [37].

The aetiology underpinning persistent pain in people with COPD is complex and underexplored. Systemic inflammatory processes inherent in COPD are likely to provide the foundation for pain susceptibility [1,3,38]. Previous authors have proposed a range of potential mechanisms that may cause, contribute to, or maintain persistent pain in people with COPD, including altered respiratory and musculoskeletal mechanics [38], postural deviations, osteoporosis, compression fractures [39], vertebral deformation, costovertebral arthropathy, [32], central sensitisation [1,38], side effects of prolonged steroid use [38], the presence of comorbid conditions including anxiety and depression [34], and personal habitus (sex and socioeconomic factors) [33]. For people with COPD attending pulmonary rehabilitation programs, musculoskeletal conditions (e.g., arthritis, back problems, and muscle cramps) have been proposed as the commonest cause of persistent pain, with low back, trunk and neck, and lower extremities described as the most common locations [1,11,32,33,40].

Roberts et al. (2013) noted limited discussion concerning pain in the international clinical practice guidelines (CPGs) for the management of COPD [30]. Given the increasing attention over the past decade concerning the prevalence and clinical implications of persistent pain in people with COPD, we were interested to know, in national and international CPGs for the broad management of COPD, (1) how often and in which contexts pain is mentioned and (2) whether recommendations and strategies are provided around the assessment or management of pain. 

## 2. Materials and Methods 

The review protocol was developed using the preferred reporting items for systematic reviews and meta-analyses (PRISMA-P) guidelines [41] and was registered with the International Prospective Register of Systematic Reviews (PROSPERO CRD42016044103). 

### 2.1. Eligibility 

Clinical practice guidelines (CPGs) were eligible for inclusion within this review if they were (1) the most recent version for the broad management of COPD and (2) published by an authoritative medical, scientific, or government body. There were no limitations set for the language of publication. Guidelines that were focused on specific aspects of disease management such as acute COPD exacerbations, pulmonary rehabilitation, or domiciliary oxygen were excluded. The original review was undertaken during 2017 with the initial search strategy limited to publicly available CPGs published in the previous decade (January 2006 to December 2016). In late 2018, an additional search was undertaken to identify any new CPGs or updated versions of the CPGs included in the original review.

### 2.2. Information Sources and Search Strategy

The search strategy was based on a previous systematic review of COPD CPGs [42]. For the current review, the strategy was reviewed, updated, and run by a single member of the research team (G.W.). The search strategy for OVID Medline is provided in Appendix A of the Appendix A. Four electronic databases (Ovid Medline, EMbase, CiNAHL, and Scopus) were searched using search terms for the population of interest (COPD, pulmonary emphysema, or pulmonary disease chronic obstructive) and publication type (guideline, consensus, position statement, guidance, or standard). The reference lists of systematic reviews identified by the electronic database search were hand searched for eligible CPGs. Seven publicly available CPG databases were also searched (The National Institute for Health and Care Excellence, Clinical Practice Guidelines Portal, National Guideline Clearing House, Guidelines International Network, Scottish Intercollegiate Guidelines Network, Canadian Medical Association Infobase: Clinical Practice Guidelines, and the International Primary Care Respiratory Group). 

### 2.3. Document Selection

A single reviewer (G.W.) screened the English titles of references retained from the searches to determine preliminary eligibility and to remove duplicates. Two independent reviewers (G.W. and H.L.) assessed the remaining abstracts for eligibility, and subsequently, for references retained (i.e., eligible for inclusion or could not be confidently excluded), accessed and reviewed full-text versions of the CPGs. Discrepancies were resolved by discussion. For references in languages other than English, the full text was obtained and a single reviewer (G.W.) assisted a person fluent in the language to determine eligibility. 

### 2.4. Data Collection 

A data extraction template was developed *a priori* for the purpose of this study. The extraction process was pilot tested on three randomly selected CPGs by two independent reviewers (G.W. and H.L.). Following refinement of the template, two independent reviewers (G.W. and H.L.) completed the data extraction for CPGs published in English and met to resolve disagreements. For guidelines published in a language other than English, a translator fluent in the language extracted data with the assistance of a single reviewer (G.W.). Data were extracted for three domains: (1) CPG demographics (title, name of developing body, year, version, country of origin, and language of publication); (2) the frequency of mention and context of the word “pain” (verbatim) in the main body of CPG or Appendix A; and (3) the presence of specific recommendations or strategies for the assessment or management of pain. 

### 2.5. Data Analysis

Data extracted regarding guideline demographics and the frequency and context of the mention of “pain” in CPGs were collated and reported descriptively. Where “pain” was mentioned in CPGs, the specific contexts were synthesized into common categories, including pain assessment and management. Findings were tabulated for comparison across guidelines. 

## 3. Results

The initial electronic database search identified 596 unique references after removal of duplicates with 44 CPGs eligible for inclusion (Figure 2). Despite meeting eligibility criteria, two CPGs were unable to be accessed due to geographic restrictions for purchase outside the United States and Japan [43,44] and a single CPG was unable to be translated from Thai [45]. Therefore, a total of 41 CPGs were included in this review, 20 published in English and 21 in a language other than English (Figure 2). The complete list of included CPGs is available in Appendix A of Appendix A. 

### 3.1. Frequency and Context of the Mention of “Pain” 

Of the 41 CPGs, 16 CPGs did not mention the term pain (39%). Of the 25 (61%) CPGs that mentioned “pain”, there were a total of 67 mentions, ranging from one mention in 10 CPGs to 10 mentions in a single guideline [46]. The verbatim mention of “pain” in CPGs is presented in Appendix A of the Appendix A. 

Mentions of “pain” in CPGs were most commonly in the context of adverse side effects of pharmacotherapies (22 mentions in 13 (32%) CPGs): phosphodiesterase-4 inhibitors (mentions were *n* = 10), smoking cessation/nicotine replacement therapies (*n* = 8), methylxanthines (*n* = 2), xanthines (*n* = 1), or beta_2_ adrenergic receptor agonists (*n* = 1) (Table 1). Other mentions of “pain” were in the context of recommendations for pain assessment (18 mentions in 10 (24%) CPGs) or management (13 mentions in 11 (27%) CPGs) or of indications/contraindications for respiratory function assessments (8 mentions in 5 (12%) CPGs) or were listed as a symptom of COPD (4 mentions in 4 (10%) CPGs) or as a complication (bullae and fractures) of COPD (2 mentions in 2 (5%) CPGs). 

Four guidelines specifically made reference to the commonality or prevalence of pain. The Finnish guideline reported pain is a common symptom of advanced COPD in 30 to 70% of patients [47]. The German CPG reported pain, dyspnoea, and fatigue to be the most common symptoms in patients in the year before death [48]. The Norway CPG stated that pain is a distressing symptom in advanced COPD [49]. The Dutch CPG reported people with COPD believe they will experience less pain than people with other chronic conditions [46]. 

### 3.2. Recommendations and Strategies Provided Around Pain Assessment

Ten (24%) CPGs provided recommendations around the assessment of pain (Table 1). Strategies recommended within CPGs for pain assessment were (1) physician to assess pain as part of a differential diagnosis, specific to the assessment of chest pain (mentions were *n* = 12 in 9 CPGs) or generalised pain as a symptom of anxiety disorder (*n* = 1); (2) physician to assess for thoracic pain as part of screening for pulmonary rehabilitation (*n* = 1); (3) self-report questionnaires for problematic sputum causing painful cough (*n* = 1); (4) the Borg rating scale as a tool to assess pain in a non-specific context (*n* = 1); and (5) cardiopulmonary exercise tests to differentiate the cause of exercise limitations (*n* = 1). 

### 3.3. Recommendations and Strategies Provided Around Pain Management 

Eleven (27%) CPGs provided recommendations around pain management (Table 1). Strategies recommended within CPGs for pain management were (1) palliative care (mentions were *n* = 5 in 5 CPGs), (2) pharmacotherapies: opioids for palliative care (*n* = 2 in 2 CPGs) or epidural analgesics for postoperative care (*n* = 1), (3) surgery (*n* = 1), (4) patient education for preoperative care (*n* = 1), (5) positive expiratory pressure (PEP) therapy as a strategy to increase ventilation for people with pain (*n* = 1), or (6) non-specific reference in the general management of COPD (*n* = 1) or around opioid dosage (*n* = 1).

### 3.4. CPGs’ Updated or New Since 2017—Has Anything Changed with Respect to Pain? 

Nine CPGs (22%) included in the original sampling frame (2006–2016) have been updated (Appendix A, Appendix A) [50,51,52,53,54,55,56,57,58], and one new CPG was identified [59]. Within updated or newly identified CPGs, there were nine mentions of “pain”. In line with the original review, “pain” was most commonly mentioned in the context of adverse side effects of pharmacotherapies (mentions, *n* = 7): phosphodiesterase-4 inhibitors (*n* = 4), nicotine replacement therapies (*n* = 2), or methylxanthines (*n* = 1). A single CPG mentioned pain in the context of assessment (physicians to assess chest pain as part of a differential diagnosis) [51], and one guideline mentioned pain associated with various organs as a common symptom in the year before death [52]. 

## 4. Discussion

Clinical practice guidelines provide recommendations and strategies for the management of chronic diseases based on the best available evidence or consensus of expert opinion. This systematic review identified 41 national and international CPGs for the management of COPD published between 2006–2016, of which, nine had been further updated by the end of 2018 and one new CPG had been developed. Of the original 41 CPGs, 16 (39%) did not mention the term “pain”. When pain was mentioned (CPGs *n* = 25, 61%; mentions *n* = 67), this was most frequently in the context of adverse reactions to specific pharmacotherapies (22 mentions in 13 (33%) CPGs). Few CPGs provided recommendations or strategies around pain assessment (18 mentions in 10 (24%) CPGs) or management (13 mentions in 11 (27%) CPGs) for people with COPD. When recommendations or strategies were provided, these were generally concerned with a differential diagnosis (10 of 10 CPGs providing pain assessment recommendations) or management in advanced COPD or palliative care (6 of 11 CPGs providing pain management recommendations). There were no additional recommendations or strategies specific to pain assessment or management provided in updated or newly developed CPGs. 

### 4.1. Given the Reported High Prevalence of Persistent Pain in COPD, Why Do So Few CPGs Address This Symptom Beyond Side Effects of Medication, Differential Diagnosis, or Palliative Management?

It could be expected that CPGs for the broad management of COPD prioritise assessment and intervention approaches that have been proven to alter the clinical course of the disease (i.e., slowing the lung function rate of decline and preventing exacerbations) or palliating disease-related symptoms [50]. As a sequelae of the underlying pathology in COPD, respiratory signs and symptoms such as chronic progressive breathlessness, cough, and sputum production are common [50]. Consequently, CPGs for COPD may be simply and appropriately reflecting the most prevalent symptoms. For example, while estimates vary, in studies that report the frequency of both chronic breathlessness and pain in people with COPD, breathlessness is more prevalent than pain (1) in clinically stable outpatients with COPD (moderate dyspnoea prevalence 94.3% versus moderate pain prevalence 32.4% [9]), (2) in COPD participants of pulmonary rehabilitations programs (dyspnoea prevalence 93% versus pain prevalence 74% [20]), and (3) in end of life/palliative care settings (“very much” breathlessness 60% versus 25% “very much” pain [36]; breathlessness 90–95% versus pain 34–77% [83]). 

Alternatively, in an environment of ever-increasing scientific literature, the development and update of CPGs is a costly undertaking. In proposing future directions for national guidelines for COPD management in Europe, Miravitlles et al. eloquently described the challenges facing developers of CPGs [84]. Apart from time, funding, and labour, developers are required to balance decisions concerning essential and non-essential CPG content (especially in CPGs planned for single chronic conditions where multiple comorbidities exist), frameworks for the choice and rating of evidence (quantitative hierarchies versus consensus opinion), stakeholder representation within CPG development, intended target audience, and strategies for the effective dissemination of CPGs. 

A key factor determining whether content is essential for inclusion within CPGs is the availability of supporting evidence at the time of the guideline development [85]. Despite the steady growth in studies reporting pain prevalence in people with COPD, few studies explore pain aetiology in this population and almost none explore pain management strategies beyond palliative or end of life care. Similar to the stated inability of van Dam van Isselt et al. (2014) to identify a single study reporting a specific intervention strategy to manage pain in people with COPD [3], we also failed to identify such studies. However, the rehabilitation community is clearly working towards this through the confirmation of appropriate pain assessment instruments [86] and qualitative explorations of the pain experiences in people with COPD in order to develop a pain management program specific to this population [87,88]. 

Perhaps a surprising finding of this review, however, was the small number of CPGs that reported the prevalence of chronic pain in COPD or listed pain as a common comorbidity. The sole CPG to report specific pain prevalence in COPD (Finland 2014 [47]) did so in the context of pain being one of the most common symptoms of advanced COPD (prevalence of pain 30–70% in the section Palliative Care). This absence might be partially explained by the time frames for publication of pain prevalence studies and lead-time requirements for the development and publication of COPD CPGs. By the end of 2018, over 30 studies had been published confirming high pain prevalence in people with COPD (Figure 3). Of these, seven were published prior to the sampling frame for this review (2006) [4,7,8,21,27,28,36]. At best, by the midpoint of this review (2010), 11 studies reporting on pain prevalence in this population would have been available to developers of the six CPGs published between 2006–2009 [48,80,81,82,89,90,91]. By 2016, there were 26 papers available as a potential evidence base to the 30 CPGs published between 2010–2015 [46,47,62,63,64,65,66,67,68,69,70,71,72,73,74,75,76,77,78,79,92,93,94,95,96,97,98,99,100,101,102]. Across 2017 to 2018, at least another nine studies have been published providing estimates of pain prevalence in people with COPD, which would be available to developers of future CPGs or of CPGs currently underway or being updated [6,19,20,26,31,32,33,34,35]. 

While the evidence base supporting chronic pain as a comorbidity in people with COPD is relatively small (but growing), there are various other factors that may contribute to the under recognition of this symptom. In the older general population, of which people with COPD form a subgroup, chronic pain, especially non-cancer related pain, is often under reported, assessed, and managed [103]. This has been suggested to reflect the deficiencies in health professional education concerning chronic pain [104], a “side-lining” of pain management as a consequence of the current focus on the opioid epidemic [104], the ongoing tensions regarding the use of opioids to palliate dyspnoea in people with respiratory disease [105,106], and/or the persistence of erroneous beliefs that pain is an expected or natural consequence of aging or will inevitably worsen over time or that putting up with pain (stoicism) will result in better pain tolerance [107]. 

### 4.2 Beyond Prevalence: Could a Case be Made for Considering Pain within CPGs for Management of People with COPD?

There is compelling evidence that in people with COPD, (1) persistent pain is more common than in people of similar age/sex in the general population [11,30,34]; (2) the risk of chronic non-cancer pain is increased [29]; and (3) for many, the pain severity should warrant not only assessment but also palliation. In a recent cross-sectional population-based analysis of Norwegians with COPD (*n* = 1199); without COPD but with arthritis (*n* = 8582), heart disease (*n* = 4109), or diabetes (*n* = 1254); or without any disease (*n* = 18,811), over half the COPD sample (56.9%) rated pain severity in the preceding four weeks as moderate to very severe, which was twice the amount of people without disease (29.5%) and second only to people with arthritis (71.9%) [34]. 

In people with COPD, the presence and severity of chronic pain has significant clinical implications. Compared to people with COPD without chronic pain, those with chronic pain have been reported to have more than double the total direct medical costs [30]. Pain has shown to be associated with poorer quality of life [108]; increased breathlessness, fatigue [2,26], and depression [35]; and impaired sleep quality [20]. Furthermore, in a population where low levels of physical activity and sedentarism are common [109,110], physical activity levels are negatively associated with pain severity [35,111]. People with COPD who live with chronic pain are less physically active than similarly aged people without COPD as well as their peers with COPD and no pain [35,111]. For this population, an inability to participate in evidence-based interventions such as pulmonary rehabilitation could be anticipated and may provide a partial explanation for the less than optimal uptake and completion rates. Whether palliating chronic pain in people with COPD leads to improvements in quality of life, symptom burden, depression, and/or participation in physical activity, however, remains to be explored. 

### 4.3. Strengths and Limitations 

This systematic review was strengthened by the comprehensive search strategy used and by not setting limitations for the publication language. Over half (21 of 41 CPGs) of included CPGs were published in a language other than English. A limitation of this review, however, is that the translation of these CPGs was not done via the back-translation method, which is the recommended method when translating documents between languages. This review was further limited by extracting only data on mentions of “pain” and not relevant synonyms. This may have missed some information around recommendations and strategies focused on pain, particularly in non-English CPGs. 

## 5. Conclusions

Chronic pain is common in people with COPD and adversely impacts quality of life, mood, breathlessness, and participation in physical activity and activities of daily living. Yet, few CPGs address the symptom of pain. The impact of effective pain management on disease trajectory, symptom burden, or quality of life in people with COPD is currently unknown. Consequently, CPG developers would need to base treatment recommendations on expert consensus or to default to CPGs created for the assessment or management of chronic pain developed for the general population. It has been estimated that there were 384 million COPD cases in 2010 [50], which is greater than the current 2018 estimate for the total population of the United States of America. Based on this 2010 estimate, if there was a country inhabited only by people with COPD and using the most conservative estimate of pain prevalence in this group (28%) [21], the number of inhabitants living with chronic pain equates to a population of at least 107,520,000: a megacity of chronic pain. From a prevalence point of view, at least, is there now enough evidence to warrant the inclusion of chronic pain within CPGs for COPD as a common clinical issue for people with COPD?

## Figures and Tables

**Figure 1 healthcare-07-00015-f001:**
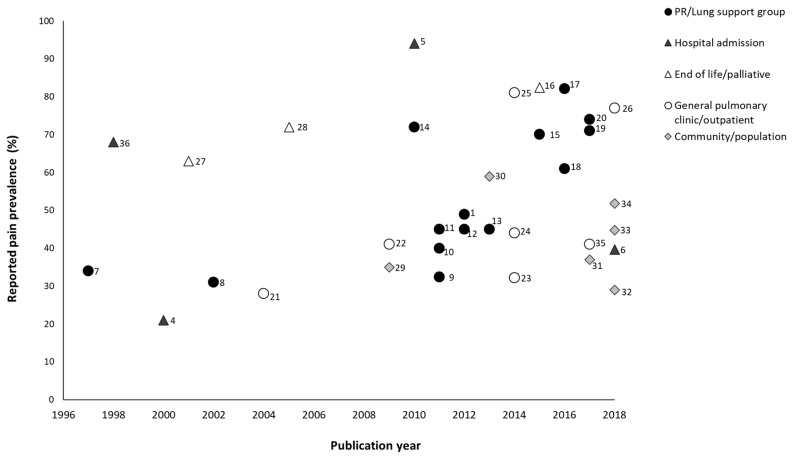
Pain prevalence reported within studies by the clinical population. PR, pulmonary rehabilitation.

**Figure 2 healthcare-07-00015-f002:**
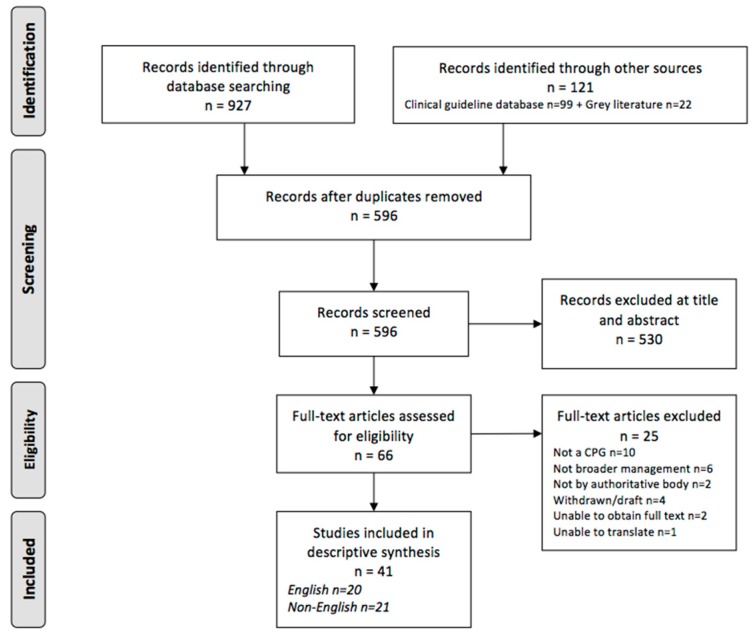
The outcome of the search strategy leading to COPD clinical practice guidelines (CPG) eligible for this review.

**Figure 3 healthcare-07-00015-f003:**
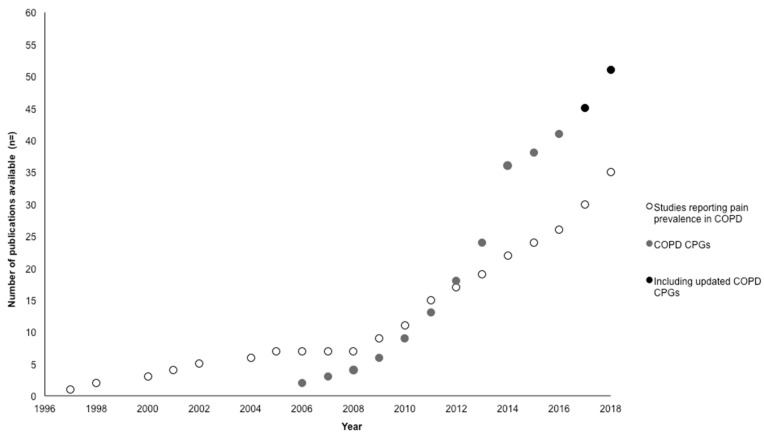
Publication of studies reporting pain prevalence in people with COPD versus COPD clinical practice guidelines (CPGs) included in this review and update.

**Table 1 healthcare-07-00015-t001:** Mentions of “pain” in specific contexts within the clinical practice guidelines for the management of COPD.

Guideline	Mentions *n* =	Symptom		Assessment			Management						Drug Effect		Lung Function
				Physician DxDx	Question/Scale	CPET	Palliative Care	Pharma	Surgery	Education	PEP	NS			
*Mentions in context (n =)*		6 *(UD+1)*		14 *(UD+1)*	3	1	5	3	1	1	1	2	22 *(UD+7)*		8
GOLD: International [60]	1												1		
Australia and NZ [51,61]	4			*1^*			1						2		1
NHG: Netherlands [62]	2			2											
Saudi Arabia [63]	1												1		
India [64]	2	1		1											
Italy [65]	1						1								
VA/DoD: USA [66]	3			1									2		
Finland [47]	3	1					1	1 *opioids*							
Poland [67]	2								1				1		
HAS: France [68]	2						1						1		
Turkey [69]	1												1		
Korea [54,70]	6												3	*1^*	3
Spain [71,72]	1			1											
Czech Republic [73]	1							1 *opioids*							
China [74]	3			1								1	1		
Chile [75]	1			1											
Ukraine [76]	2			2											
Algeria [77]	1														1
Norway [49]	9	2		1	2	1					1	1	1		
South Africa [78]	2									1*					1
United Kingdom [79]	5			2			1								2
CTS: Canada [80,81]	1												1		
Germany [48,52]	2	1	*1^*										1	*4^*	
NVALT: Netherlands [46]	10	1		2	1								6		
Malaysia [82]	1							1 *ANAL**							
UD BC: Canada [55]	1												*1^*		
New Singapore [59]	1												*1^*		

* Analgesic epidural specific to pre-/post-surgery; ^ pain mention in update of CPG; CPET, cardiopulmonary exercise testing; DxDx, differential diagnosis; PEP, positive expiratory pressure therapy; Pharma, pharmacological; NS, non-specific; Question, questionnaire; and UD, updated CPG.

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
