# Peer review of "Systematic Review of Pain in Clinical Practice Guidelines for Management of COPD: A Case for Including Chronic Pain?"

_healthcare, 2019, doi:10.3390/healthcare7010015_

Round 1

Reviewer 1 Report

The manuscript titled as "Systematic review of pain in clinical practice  guidelines for management of COPD: a case for  including chronic pain?"describes describe the frequency and  of contexts pain in clinical  practice guidelines (CPGs) for the broad management of COPD.

The manuscript is unnecessarily long and needs to be shortened.

The discussion should focus on the interpretation of the results and conclusion should be shortened as it appears to be more summary than conclusions.

Author Response

Reviewer 1 

The manuscript titled as "Systematic review of pain in clinical practice guidelines for management of COPD: a case for including chronic pain?" describes the frequency and of contexts pain in clinical practice guidelines (CPGs) for the broad management of COPD. 

The manuscript is unnecessarily long and needs to be shortened. The discussion should focus on the interpretation of the results and conclusion should be shortened as it appears to be more summary than conclusions.

Authors’ response.

In order to meet the reporting requirements for systematic reviews (PRISMA) the majority of journals suggest a word limit of 5000 words (our original manuscript was ~3500). We have reviewed the manuscript and edited where possible (Methods) and Discussion section. We have removed ‘summary’ information from the conclusion section.

Reviewer 2 Report

I thought this was an excellent study with excellent methodology.

The authors differentiated the different types of pain patients with COPD may have (acute illness, persistent pain, related to anxiety and palliative).

The graphs and tables were well set out.

I suggest the authors also add osteoporosis, crush fractures and side effects of steroids as potential causes for pain.

The authors do not the impact of the "opioid crisis" on paitents not receiving pain management appropriatley, but perhaps the authors could acknowledge the tension of giving opioids to patients with respiratory failure, and the fear of worsening respiratory failure.  This paper BMJ. 2003 Sep 6;327(7414):523-8 suggests that this fear should not be valid.

Author Response

Reviewer 2

Thought this was an excellent study with excellent methodology. The authors differentiated the different types of pain patients with COPD may have (acute illness, persistent pain, related to anxiety and palliative). The graphs and tables were well set out.

I suggest the authors also add osteoporosis, crush fractures and side effects of steroids as potential causes for pain.

The authors do not (sic) the impact of the "opioid crisis" on patients not receiving pain management appropriately, but perhaps the authors could acknowledge the tension of giving opioids to patients with respiratory failure, and the fear of worsening respiratory failure.  This paper BMJ. 2003 Sep 6;327(7414):523-8 suggests that this fear should not be valid.

Authors’ response

Potential causes of pain- 

We have added osteoporosis and crush (compression) fractures (Ref 39) -Introduction Line 67 and side effect of prolonged steroid use (Ref 38) -Introduction line 68-69

Opioids to respiratory failure patients – 

We have added the suggested point and reference into the relevant section of the Discussion (line 284).

Reviewer 3 Report

This is a nice survey study of published articles focusing on clinical issues (pain) of patients with COPD. Whereas the staticital / clinical investigations are fine, the influence of diagnostic quality /specificity and error rate is not mentioned. At least a short chapter of error rates in COPD (for example unknown or not investigated focal stenosis, history of smoking, degree of emphysema should be included in order to estimate pain - associated circumstances and the potential diagnostic error rates of investigated cases. A useful article, which should be included in the references is: KUNZE, Klaus Dietmar. Limits of Morphological Diagnostics. Diagnostic Pathology, [S.l.], v. 2, n. 1, feb. 2016. ISSN 2364-4893. Available at: <http://www.diagnosticpathology.eu/content/index.php/dpath/article/view/103>.
This would improve the article's quality to a high degree.

Author Response

Reviewer 3

This is a nice survey study of published articles focusing on clinical issues (pain) of patients with COPD. Whereas the statistical / clinical investigations are fine, the influence of diagnostic quality /specificity and error rate is not mentioned. At least a short chapter of error rates in COPD (for example unknown or not investigated focal stenosis, history of smoking, degree of emphysema should be included in order to estimate pain - associated circumstances and the potential diagnostic error rates of investigated cases. A useful article, which should be included in the references is: KUNZE, Klaus Dietmar. Limits of Morphological Diagnostics. Diagnostic Pathology, [S.l.], v. 2, n. 1, feb. 2016. ISSN 2364-4893. Available at: <http://www.diagnosticpathology.eu/content/index.php/dpath/article/view/103>.

This would improve the article's quality to a high degree. 

Authors’ response

We have interpreted this comment to be in the context of diagnosis of COPD (under or over estimates of COPD prevalence, underdiagnosis based on the current GOLD criteria (spirometry/phenotypes)) and whether pain prevalence is impacted by how COPD is diagnosed. 

If this is the point, this would be beyond the scope of this review as we purposively reviewed clinical practical guidelines for management of COPD to identify if and how ‘pain’ is reflected. If we had undertaken a systematic review of studies reporting pain prevalence in people with a diagnosis of COPD, then this would be a very pertinent point ( i.e., how COPD is diagnosed will have an impact of prevalence rates).  

Round 2

Reviewer 3 Report

The authors did not respond correctly to the comments of the reviewer. Especially, the mandatory request to comment the limitations of such study have been completely disregarded. I repeat:

Whereas the statistical / clinical investigations are fine, the influence of diagnostic quality /specificity and error rate is not mentioned. At least a short chapter of error rates in COPD (for example unknown or not investigated focal stenosis, history of smoking, degree of emphysema should be included in order to estimate pain - associated circumstances and the potential diagnostic error rates of investigated cases. A useful article, which should be included in the references is: KUNZE, Klaus Dietmar. Limits of Morphological Diagnostics. Diagnostic Pathology, [S.l.], v. 2, n. 1, feb. 2016. ISSN 2364-4893. Available at: <http://www.diagnosticpathology.eu/content/index.php/dpath/article/view/103>.

This would improve the article's quality to a high degree and avoid any incorrect conclusions.